# Idiopathic inflammatory myopathy human derived cells retain their ability to increase mitochondrial function

Carla Basualto-Alarcón[1,2,3], Félix A. Urra[2,4], María Francisca Bozán[5], Fabián Jaña [3], Alejandra Trangulao[6,7], Jorge A. Bevilacqua [1,6,7], J. César Cárdenas [2,8,9,10]*

1 Anatomy and Legal Medicine Department, Faculty of Medicine, University of Chile, Santiago, Chile, 2 Geroscience Center for Brain Health and Metabolism, Santiago, Chile, 3 Health Sciences Department, Universidad de Aysén, Coyhaique, Chile, 4 Molecular and Clinical Pharmacology Program, Institute of Biomedical Sciences, Faculty of Medicine, University of Chile, Santiago, Chile, 5 Department of Internal Medicine, Hospital Clínico Universidad de Chile; Faculty of Medicine, University of Chile, Santiago, Chile, 6 Department of Neurology and Neurosurgery, Hospital Clínico Universidad de Chile; Faculty of Medicine, University of Chile, Santiago, Chile, 7 Department of Neurology and Neurosurgery, Clínica Dávila, Santiago, Chile, 8 Laboratory of Cellular Metabolism and Bioenergetics, Center for Integrative Biology, Faculty of Sciences, Universidad Mayor, Santiago, Chile, 9 The Buck Institute for Research on Aging, Novato, California, United States of America, 10 Department of Chemistry and Biochemistry, University of California, Santa Barbara, California, United States of America

* julio.cardenas@umayor.cl

**Data Availability Statement:** All relevant data are within the manuscript.

**Funding:** This research was supported by the following grants: FONDECYT postdoctoral

## Abstract

Idiopathic Inflammatory Myopathies (IIMs) have been studied within the framework of auto-immune diseases where skeletal muscle appears to have a passive role in the illness. However, persiting weakness even after resolving inflammation raises questions about the role that skeletal muscle plays by itself in these diseases. "Non-immune mediated" hypotheses have arisen to consider inner skeletal muscle cell processes as trigger factors in the clinical manifestations of IIMs. Alterations in oxidative phosphorylation, ATP production, calcium handling, autophagy, endoplasmic reticulum stress, among others, have been proposed as alternative cellular pathophysiological mechanisms. In this study, we used skeletal muscle-derived cells, from healthy controls and IIM patients to determine mitochondrial function and mitochondrial ability to adapt to a metabolic stress when deprived of glucose. We hypothesized that mitochondria would be dysfunctional in IIM samples, which was partially true in normal glucose rich growing medium as determined by oxygen consumption rate. However, in the glucose-free and galactose supplemented condition, a medium that forced mitochondria to function, IIM cells increased their respiration, reaching values matching normal derived cells. Unexpectedly, cell death significantly increased in IIM cells under this condition. Our findings show that mitochondria in IIM is functional and the decrease respiration observed is part of an adaptative response to improve survival. The increased metabolic function obtained after forcing IIM cells to rely on mitochondrial synthesized ATP is detrimental to the cell's viability. Thus, therapeutic interventions that activate mitochondria, could be detrimental in IIM cell physiology, and must be avoided in patients with IIM.

#3150623 (CB), FONDECYT postdoctoral #3170813 (FU), FONDECYT #11170291 (FJ), FONDECYT #1160332 (CC), FONDECYT #1151383 (JB), and ANID/FONDAP/15150012 (CC). https://www.anid.cl/ The funders had no role in study design, data collection and analysis, decision to publish, or preparation of the manuscript.

**Competing interests:** The authors have declared that no competing interests exist.

## 1. Introduction

Idiopathic Inflammatory Myopathies (IIMs) are the most frequent acquired myopathies observed in clinical practice [1] and represent a heterogeneous group of chronic, subacute, or acute acquired muscle disorders of unknown etiology. They share the common feature of muscle inflammation which leads to generalized weakness. In addition, other organs are frequently involved in IIMs (e.g. skin, joints, lungs, gastrointestinal tract, heart, etc.) and these conditions greatly contribute to morbidity and mortality.

According to the recent classification criteria that consider clinical, histopathological and myositis specific antibodies, IIMs may be subdivided in at least five main subtypes: dermatomyositis (DM), polymyositis (PM), immune- mediated necrotizing myopathy (IMNM), overlap myositis (OM), and sporadic inclusion body myositis (sIBM) [1].

Despite the presence of inflammation as a common feature in IIMs, the underlying pathophysiological mechanisms that determine the wide range of manifestations in IIMs are yet to be fully understood. It is generally hypothesized that the obvious consequence of inflammation must be myofiber damage. However, a growing body of evidence supports the view that other mechanisms might also be involved in the pathogenesis of IIM. This evidence has led some to argue that the cause of muscle dysfunction is far more complex than just immune- mediated inflammation [2]. In fact, a lack of correlation between inflammation and skeletal muscle weakness has been reported [3], suggesting that different intracellular processes can also play a role in IIMs. Furthermore, not all patients have a positive response to immunosuppression, yielding a miscorrelation between inflammation and clinical response [2]. As a result, several non-immune-mediated alterations have been proposed to participate in the pathogenesis of PM and DM muscle cells such as disturbances in oxidative phosphorylation (OXPHOS), impairment of ATP production, altered calcium handling, autophagy, and the unfolded protein response [2–5]. Histological and histochemical abnormalities in muscle biopsies from both DM and PM patients have suggested mitochondrial damage at different levels [6, 7]. An abnormal activity of respiratory complexes I, II, III, IV and the citrate synthase has been shown in muscle homogenates from PM patients [8]. In addition, dysfunction of cytochrome c oxidase (COXc) and succinate dehydrogenase (SDH) has also been described in biopsies from IIM patients [6, 7]. *In vivo* measurements of muscle metabolism using phosphorus magnetic resonance spectroscopy ($^{31}$P-MRS) showed decreased levels of phospho-creatine (PCr), ATP and elevated inorganic phosphate (Pi)/PCr ratios during rest, exercise, and recovery. Altered resting values and slowed recovery of high energy phosphates after exercise was interpreted as abnormal mitochondrial function [9, 10]. In fact, impaired mitochondrial respiration rates in DM and PM were described by Cea et al (2002), Newman et al. (1992) [11, 12] and Pfleiderer (2004) [13] by using this imaging technique. Interestingly, Cea and Pfleiderer proposed an impaired blood supply as the main cause for the diminished oxidative metabolism observed in DM and PM patients rather than primary mitochondrial abnormalities. The marked muscle atrophy and impaired muscle function seen in DM and PM patients, even after treatment, have also been associated with a hypothetical metabolic dysfunction [14].

Drawing on current evidence, one could infer that an altered mitochondrial profile in IIMs should be expected. However, to the best of our knowledge, no real time measurement toward determining mitochondrial function in these patients seems to be available. Here, mitochondrial function, metabolic flexibility and its potential role in determining IIM cell viability were studied. We hypothesized that mitochondria from IIM-derived cells would show a diminished oxygen consumption rate (OCR), as well as diminished ATP production. We expected that a metabolic challenge imposed by a change in the carbon source in the growth medium would increase mitochondrial function, with a concomitant improvement in cellular fitness. In fact,

OCR increased in IIM derived cells reaching values similar as those observed in control cells, showing a preserved metabolic flexibility. Although, on the contrary of what was expected, the ability of IIM cells to adapt to this metabolic stress did not result in increased cellular fitness and endurance to stress.

## 2. Materials and methods

### 2.1 Reagents

All reagents were obtained from Sigma-Aldrich Corp. (St. Louis, MO, USA). Stock solutions of all compounds were prepared in dimethyl sulfoxide (DMSO). Collagenase type 1 was obtained from Worthington (Lakewood, NJ, USA). All antibodies were obtained from Dako (Glostrup, Denmark).

### 2.2 Muscle biopsy samples and pathological confirmation

Deltoid muscle biopsy specimens from five patients clinically diagnosed with IIM were taken for histological and immunohistological examination (Table 1). Controls were obtained from four age-matched patients who underwent shoulder surgery. This study was undertaken with ethical approval from the "Research Ethics Committee at the Hospital Clínico Universidad de Chile" and is in compliance with the provisions of the Declaration of Helsinki. All patients gave their written informed consent before the surgical procedure for obtaining the muscle biopsy sample.

The confirmation for Idiopathic Inflammatory Myopathy (IIM) was accomplished based on the regular analysis of the biopsy sample (see Materials and Methods). M = male; F = female; IU/L = International Units/liter; DM = dermatomyositis; IMNM = immune- mediated necrotizing myopathy.

### 2.3 Ethics statement

This study was undertaken with ethical approval from the "Research Ethics Committee at the Hospital Clínico Universidad de Chile" and is in compliance with the provisions of the Declaration of Helsinki. All patients gave their written informed consent before the surgical procedure for obtaining the muscle biopsy sample.

### 2.4 Patient consent

Written, informed consent was obtained from the individual(s) for the publication of any potentially identifiable images or data included in this article.

After excision, muscle samples were immediately frozen in isopentane previously cooled in liquid nitrogen. Samples were stored in -80˚C until biopsy processing. We studied 10 µm tissue sections employing the following techniques: hematoxylin eosin, Gomori trichome, PAS, Oil Red O, NADH, SDH, COX, ATPase 9.4, 4.6, and 4.3. Antibodies were used in the following dilutions: HLA I (1:2000), HLA II (1:500), C5b9 (1:50), and CD68 (1:50).

**Table 1. Patient clinical data.**

| Patient | Age | Gender | Weakness evolution | Total Creatine Kinase (IU/L) | Corticoid use before biopsy | Diagnosis |
|---------|-----|--------|--------------------|-----------------------------|----------------------------|-----------|
| 101 | 59 | M | 6 months | 2079 | No | DM |
| 103 | 63 | M | 3 weeks | 3969 | 2 days | DM |
| 104 | 70 | F | 6 months | 8799 | 2 days | IMNM |
| 105 | 70 | M | 1 month | 2395 | No | IMNM |
| 109 | 63 | M | 12 months | 6331 | No | IMNM/DM |

## 2.5 Cell cultures

Human primary culture cells were isolated from fresh biopsy samples obtained from the deltoid muscle of IIM patients and age-matched controls, following the guidelines of the research ethics committee at the Hospital Clínico Universidad de Chile. After extracting a biopsy, muscle tissue was immediately placed in standard culture medium (high glucose DMEM + F12; 10% fetal bovine serum). Then, skeletal muscle tissue was mechanically disaggregated and subjected to collagenase treatment for 30 min (0.2% Collagenase in PBS), under gentle agitation. The suspension was spun down at 2500 RPM for 10 min. The resulting pellet was washed with 4 mL of PBS and the supernatant was centrifuged again (2500 RPM for 10 min). Finally, the cellular pellet that included myoblasts and myofibroblasts was plated at 30% - 50% confluence and grown until reaching 80% confluence. Control- and IIM-derived cells were grown in standard culture medium: Dulbecco´s modified Eagle´s medium (DMEM), containing 25 mM glucose and 4 mM glutamine supplemented with 10% fetal bovine serum (FBS), penicillin (100 IU/mL), and streptomycin (100 μg/mL).

## 2.6 Cell culture conditions for metabolic stress

For the generation of cellular subpopulations with different metabolic phenotypes, a fraction of control and IIM-derived cells were maintained in standard culture media (25 mM glucose and 4 mM glutamine). Other fraction of cells was grown in media in which glucose was replaced by 10 mM galactose. In both conditions, cells were maintained in a humidified atmosphere at 37°C and 5% $CO_2$.

## 2.7 Real-time metabolic analysis

For oxygen consumption measurements, control and IIM-derived cells were studied with an Extracellular Flux Analyzer (Seahorse XF$^e$96; Agilent Technologies™, CA, USA). Briefly, Control and IIM–derived cells were trypsinized and pre-plated for 30 min to enriched cell suspension with myoblasts. Then, 20.000 cells were seeded in each well and the experiment was performed after 24 hrs. (growing conditions: 37°C in 5% $CO_2$). One hour before running the experiment, culture medium was replaced with "assay medium" (DMEM, 1 mM glutaMAX®, 10 mM glucose, pH 7.4). Three points were measured to establish the baseline of the oxygen consumption rate (OCR), and after the sequential injection of oligomycin [1μM], FCCP [0.1 μM] and rotenone+antimycin A [1 μM each]. This protocol allowed us to reveal basal, maximal, ATP-coupled and proton leak-linked respiration. All data were normalized by cell number.

## 2.8 Intracellular ATP determination

ATP levels were determined with CellTiter-Glo Luminescent Cell Viability Assay kit (Promega, USA) according to the manufacturer's specifications. Control and IIM-derived cells ($1x10^5$ cells/mL) were seeded into 96-well plates and grown overnight. To determine the contribution of OXPHOS on total ATP levels, cells were incubated for two hours with 2 μM oligomycin. After exposure, the cells were washed twice with cold-PBS to remove the culture medium and re-suspended in 20 μL PBS.

## 2.9 Mitochondrial membrane potential (ΔΨm) determination

ΔΨm in Control and IIM-derived cells was determined by flow cytometry using the potentiometric probe tetramethylrhodamine methyl ester (TMRM, Molecular Probe) in non-quenching mode. Cells (1.5 x $10^5$ cells/mL) were treated with Dimethyl sulfoxide (DMSO) or FCCP

(0.5 and 1 μM), which was used as positive control, for 30 min. Then, cells were incubated with 5 nM TMRM for 20 min, washed with cold-PBS, collected, re-suspended and fluorescence measured using a FACS Calibur flow cytometer.

## 2.10 Determination of intracellular Reactive Oxygen Species (ROS) measurements

The generation of intracellular oxidative stress was determined using the dihydroethidium (DHE) probe. Control- and IIM-derived cells were grown in GLU and GAL media, seeded in 12-well plates and allowed overnight to attach. Then, culture media was replaced by a solution containing 5 μM DHE in HBSS and incubated for 20 min in the dark. Afterward, cells were washed, trypsinized, and resuspended in 200 μL of HBSS and measured by FACS Calibur flow cytometer.

## 2.11 Cell viability assay

Cell viability was evaluated using the ability of live cells to exclude propidium iodide (PI). Control- and IIM-derived cells were submitted to different challenges in 24-well plates treated with PI, collected, washed and run in a FACS Calibur flow cytometer (Becton-Dickinson, San Jose, CA) for quantification of PI incorporation. For "redox stress" conditions, cells were pre-incubated with hydrogen peroxide ($H_2O_2$) as an oxidant stimulus [100 uM], 48 hours before reading the experiment.

For comparison, we show the "Δ death cell %", that corresponds to the difference in death percentage in the "experimental medium" (galactose, GAL), minus the percentage of death in "regular medium" (glucose, GLU): Δ = % death in galactose medium—% death glucose medium. In this condition, positive values for Δ death cell % represent cells more prone to die in GAL than in GLU media, whereas negative values represent cells more likely to survive in GAL media than in GLU media.

## 2.12 Determination of mitochondrial mass

Control and IIM myoblasts were loaded with the cardiolipin-binding probe 10-N-nonyl acridine orange (NAO). Briefly, $1.5 \times 10^5$ cells/mL were incubated with NAO probe, at a concentration of 0.1 μM, for 15 minutes. After that, cells were washed with PBS, collected, re-suspended and fluorescence was measured by flow cytometry (FACS Calibur).

## 2.13 Statistics

All experiments were performed at least three independent times. Values were expressed as mean ± SEM. For statistical analysis, Mann-Whitney and Wilcoxon tests were performed, as well as two-way ANOVA tests with Bonferroni post-test to determine significance. Significance level was set at $p < 0.05$ value. For statistical analysis, GraphPad Prism 6 was used.

# 3. Results

## 3.1 Control and IIM-derived cell bioenergetic characterization, in high glucose media

Myotubes and myoblasts are traditionally cultured in high glucose growing media (see Materials and Methods), and our first attempts to bioenergetically characterize control and IIM-derived cells were conducted in this scenario. Real time oxygen consumption measurements, as indicators of mitochondrial respiration, revealed that in IIM-derived cells, mitochondria exhibit a significantly lower basal OCR compared to normal cells (Fig 1A and 1B). Sequential

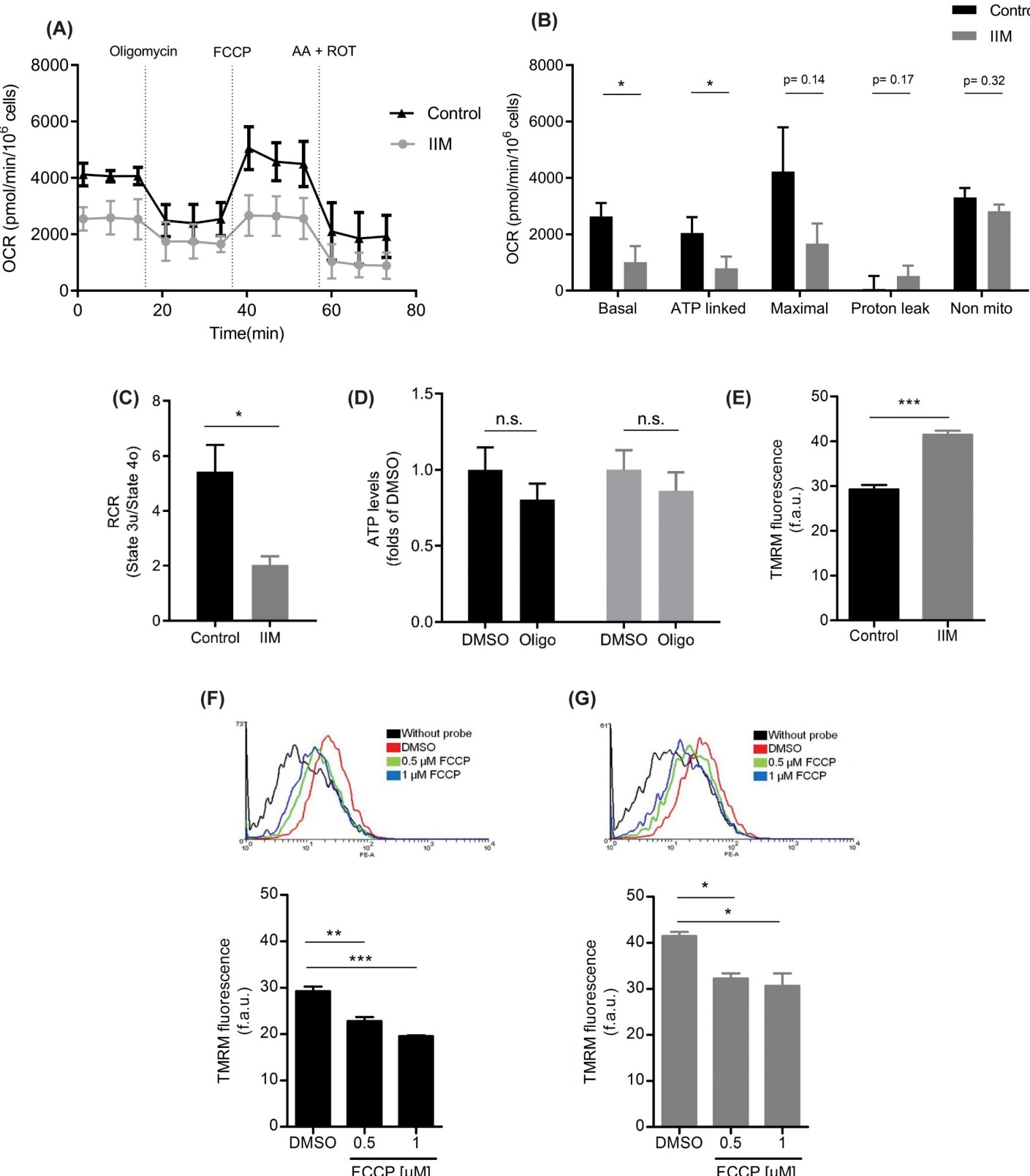

**Fig 1. IIM-derived cells show diminished oxygen consumption rate, high mitochondrial membrane potential but similar total ATP levels than control-derived cells. (A)** Representative OCR profile plot in control and IIM-derived cells. **(B)** Oxygen consumption rate (OCR) in control and IIM-derived cells from biopsy samples

showed a significant decrease of ATP-linked OCR and a marked tendency to decrease basal and maximal OCR when compared to controls. Proton leak and non-mitochondrial (non-mito) OCR exhibited no changes. **(C)** Respiratory Control Ratio (RCR) was significantly lower in IIM derived cells than in control. **(D)** Total ATP levels measured in control and IIM-derived cells showed no differences in basal conditions (DMSO) or after treatment with oligomycin (Oligo). **(F)** Mitochondrial membrane potential ($\Delta\Psi$m) determined by TMRM in non-quenching mode showed that IIM-derived cells have a higher $\Delta\Psi$m compared with control derived cells. Original FACS traces showing $\Delta\Psi$m measure in **(F)** Control and **(G)** IIM-derived cells treated with FCCP (0.5 and 1 μM) exhibiting the expected $\Delta\Psi$m depolarization. Data shown represent the mean ± SEM of three independent experiments. $^{*}p < 0.05$, $^{**}p < 0.01$, $^{***}p < 0.001$, n.s. not significant.

injections of Oligomycin, Carbonyl cyanide-4-(trifluoromethoxy) phenylhydrazone (FCCP) and Rotenone + Antimycin A, revealed that ATP-linked respiration followed the same trend (Fig 1B), with an OCR significantly lower in IIM-derived cells compared with control cells ($p < 0.05$). Interestingly, the proton leak OCR, that represents the oxygen consumption not associated with ATP generation, showed a tendency to be higher in IIM condition (Fig 1B). Consistently, the Respiratory Control Ratio (RCR), which represents the mitochondrial coupling state, was significantly lower in IIM-derived cells ($p < 0.05$) (Fig 1), suggesting an uncoupled OXPHOS. Finally, the non-mitochondrial OCR, showed no differences (Fig 1B). This last parameter showed unexpected high values in both Control and IIM conditions which may reflect the presence of a higher cellular metabolism in myoblasts.

To complement the bioenergetic analysis, measurements of total intracellular ATP levels and $\Delta\Psi$m were performed. Both control and IIM-derived cells showed similar levels of total basal ATP, as well as a minimal drop on ATP levels after ATP synthase inhibition with oligomycin (Fig 1D). This result suggests that in standard, high glucose culture conditions, both control and IIM-derived cells rely mostly on glycolysis to synthesize their ATP, as oligomycin, an inhibitor of mitochondrial ATP synthase did not decrease ATP levels as expected. Regarding the $\Delta\Psi$m, an increased TMRM incorporation was observed in IIM-derived cells compared with controls ($p < 0.001$) (Fig 1E). This finding was indicative of mitochondrial hyperpolarization that can be dissipated by two different FCCP concentrations, as well as the normal $\Delta\Psi$m in normal cells (Fig 1F and 1G).

## 3.2 IIM mitochondria are metabolically flexible and sensitive to oxidative stress

Generally, muscle derived cells are cultured in high glucose media which favors glycolytic metabolism over oxidative metabolism even in the presence of oxygen ("Crabtree effect"). Thus, to truly reveal mitochondrial functional state under a condition where mitochondrial function is a determinant of cell viability, we changed the carbohydrate availability using a culture medium with galactose (GAL) instead of glucose (GLU) (15). IIM- and control-derived cells were cultured during seven days in GAL-medium, where GAL metabolization by glycolysis yields no net ATP production, forcing cells to rely on mitochondria [15, 16]. Surprisingly, OCR measurement showed that IIM-derived cells were able to increase their mitochondrial function to the same level as normal-derived cells when grown in GAL-media. Basal OCR did not show differences between control and IIM-derived cells in GAL-media (Fig 2A). Conversely, ATP-linked OCR, which was significantly lower for IIM grown in GLU-media in comparison with the control, did increase to the same level as the control in GAL-media (Fig 2B).

Determination of total intracellular ATP levels revealed that both IIM and control-derived cells in GAL media rely on mitochondria for ATP synthesis, since the inhibition of the ATP synthase with oligomycin significantly reduced the levels of ATP (Fig 2C). Taken together, these results suggest that mitochondria retain the potential to reach full functionality in IIM cells independent of the growing culture medium, and that the decrease observed in mitochondrial function in GLU-media may correspond to an adaptive response undergoing the aforementioned "Crabtree effect". Nonylacridine orange (NAO) staining show no differences

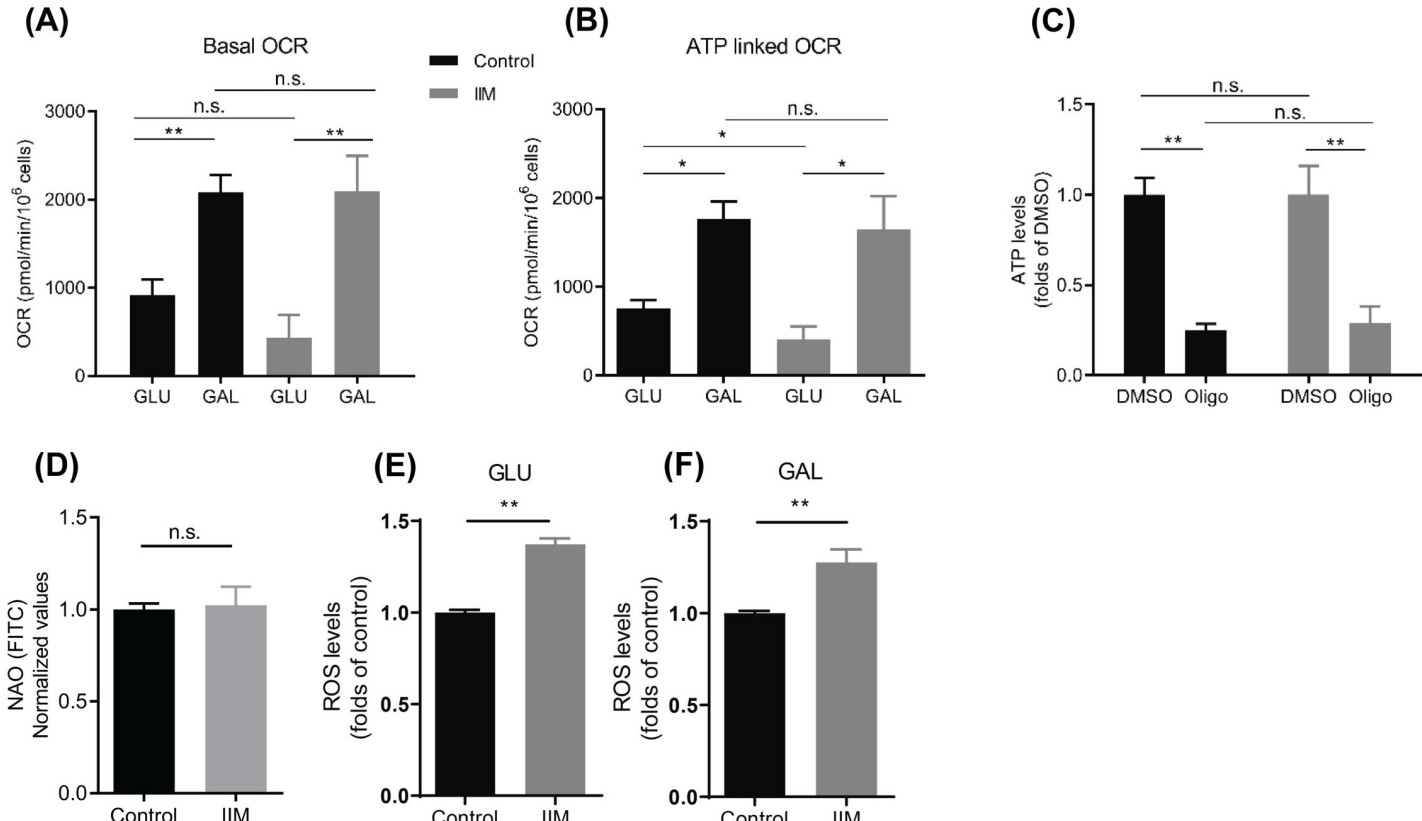

**Fig 2. A metabolic challenge reveals functional mitochondria in IIM-derived cells and a mitochondrial-dependent sensitivity to ROS-mediated cell death.** Control and IIM-derived cells were grown for seven days either in glucose (GLU) or galactose (GAL) media. Oxygen consumption rate (OCR) show that both Control and IIM-derived cells were able to; (A) increase basal and (B) ATP linked OCR when cultured in GAL. (C) Total ATP levels in control and IIM-derived cells showed negligible differences in basal conditions (DMSO) and a significant drop when treated with the ATP synthase inhibitor oligomycin (Oligo) (1 µM) for 2 h. (D) Mitochondrial mass was determined using the cardiolipin fluorescence label NAO. No differences were found between Control and IIM-derived cells after seven days in GAL media (E and F) Reactive oxygen species (ROS) levels were measured using DHE. In both growing media (GLU and GAL) IIM derived cells showed increased ROS levels when compared with control cells.

between control and IIM cells (Fig 2D) suggesting no differences in mitochondrial mass. To strengthen this point, we labeled biopsy samples from control and IIM patients with an antibody against the outer mitochondrial protein VDAC and we determined its expression by quantifying the number of pixels per area in confocal microscope images. As shown in the S1A Fig, no difference in the distribution nor in the expression were found between normal and IIM samples. In addition, using an antibody cocktail we determined the expression of the electron transport chain complexes by Western blot in biopsy samples from control and IIM patients. As shown in S1B Fig, no changes in the expression of complex I, III, IV and V were observed between control and IIM samples. Due to technical reasons complex II was not identified in any of our samples. These results strengthen the idea that no difference in mitochondrial mass exists between normal and IIM cells. More experiments are necessary to confirm this point.

Over-functioning mitochondria are a potential source of ROS, which are a known mediator of skeletal muscle cell damage [17, 18]. Thus, we decided to explore whether IIM-derived cells generate more ROS levels in basal conditions. Compared with control cells, IIM-derived cells showed elevated ROS levels in both conditions, GLU- and GAL-media (Fig 2E and 2F).

Finally, we decided to test IIM-cell sensitivity to death under an additional oxidative challenge. To this end, we determined cell viability in control and IIM-derived cells grown in GLU

and GAL-media, challenged with hydrogen peroxide ($H_2O_2$) used as an oxidant stimulus. As shown in Fig 3A, IIM-derived cells reach the highest death values in GLU and GAL culture media, both in basal and the "challenged" condition (H2O2). For comparison, we decided to show the percentage delta of cells that died in GAL minus GLU medium. This result corresponds to the "percentage of death in the GAL-media" minus the "percentage of death in GLU-media" (see Materials and Methods). As shown in Fig 3B, the metabolic stress imposed by the change from GLU to GAL-media displays no perceptible difference between control and IIM cells. Interestingly, when a redox stress (100 μM $H_2O_2$ for 48 h) was applied, IIM-derived cells displayed an increase in the delta percentage of cell death, which suggests that derived the activity of mitochondria renders IIM-derived cells more prone to oxidative damage. In contrast, control-derived cells showed negative values which in turn suggests that fewer cells die in GAL-media when mitochondria are working to supply the majority of ATP.

Taken together, our results indicate that: (1) in IIM, mitochondria retain their ability to increase oxidative function in order to meet cellular ATP demands and (2) the increase in oxidative mitochondrial function may have a detrimental effects on IIM cell viability.

## 4. Discussion

It has been proposed that in IIM, the inflammatory local response is not the only factor responsible for the pathophysiology of this illness [2]. There is evidence that points to mitochondrial malfunction also being responsible, proposing that mitochondria are more likely to produce high levels of ROS and a decrease in ATP synthesis capacity [2–12, 18]. In this work, we show for the first time that in human derived IIM myoblasts, mitochondria are not dysfunctional and retain their ability to adapt and respond to environmental signals, at a cost that renders cells more prone to death after a specific, oxidative, insult.

IIM-derived cells grown in a high glucose medium exhibited a lower OCR profile and ΔΨm hyperpolarization compared to control-derived cells. These results were not followed by decreases in ATP levels (Fig 1), probably due to the "Crabtree effect" that maintains adequate ATP levels in both types of cells under these culture conditions. The Crabtree effect is characterized by the inhibition of respiration by high concentrations of glucose or fructose. In this condition, ATP is generated primarily by glycolysis and not by mitochondria. Fig 2 shows that when we "forced" mitochondria to work by substituting galactose for glucose in the media, both control and IIM-derived mitochondria were able to increase their respiration rates. We interpret this result as a demonstration of the IIM cell ability to adapt successfully to a metabolic stress condition. Nonyl acridine orange (NAO) labeling of cells in culture, in addition to the labeling of the outer mitochondrial protein VDAC and Western blot of the electron transport chain complexes I, III, IV and V on biopsy samples from control and IIM patients, suggests that no changes in mitochondrial mass between normal and IIM samples exist. NAO binds cardiolipin, which is highly concentrated in the mitochondria independent of ΔΨm [19], however under certain circumstances it accumulates in the mitochondria in a ΔΨm-dependent fashion [20, 21]. To confirm that the mitochondrial mass is similar between control and IIM samples, further experiments measuring the mitochondrial/nuclear DNA ratio are necessary.

Interestingly, IIM-derived cells showed a tendency to die in higher proportions than controls as a result of this metabolic stress. However, we should keep in mind that this trend only reached significance after adding a redox stimulus (Fig 3). This observation suggests that higher mitochondrial metabolic rates in IIM-derived cells increase their susceptibility to die. Mechanistically, it has been proposed that activation of endoplasmic reticulum stress in IIM leads to ROS production. Such increased ROS production would occur through increased

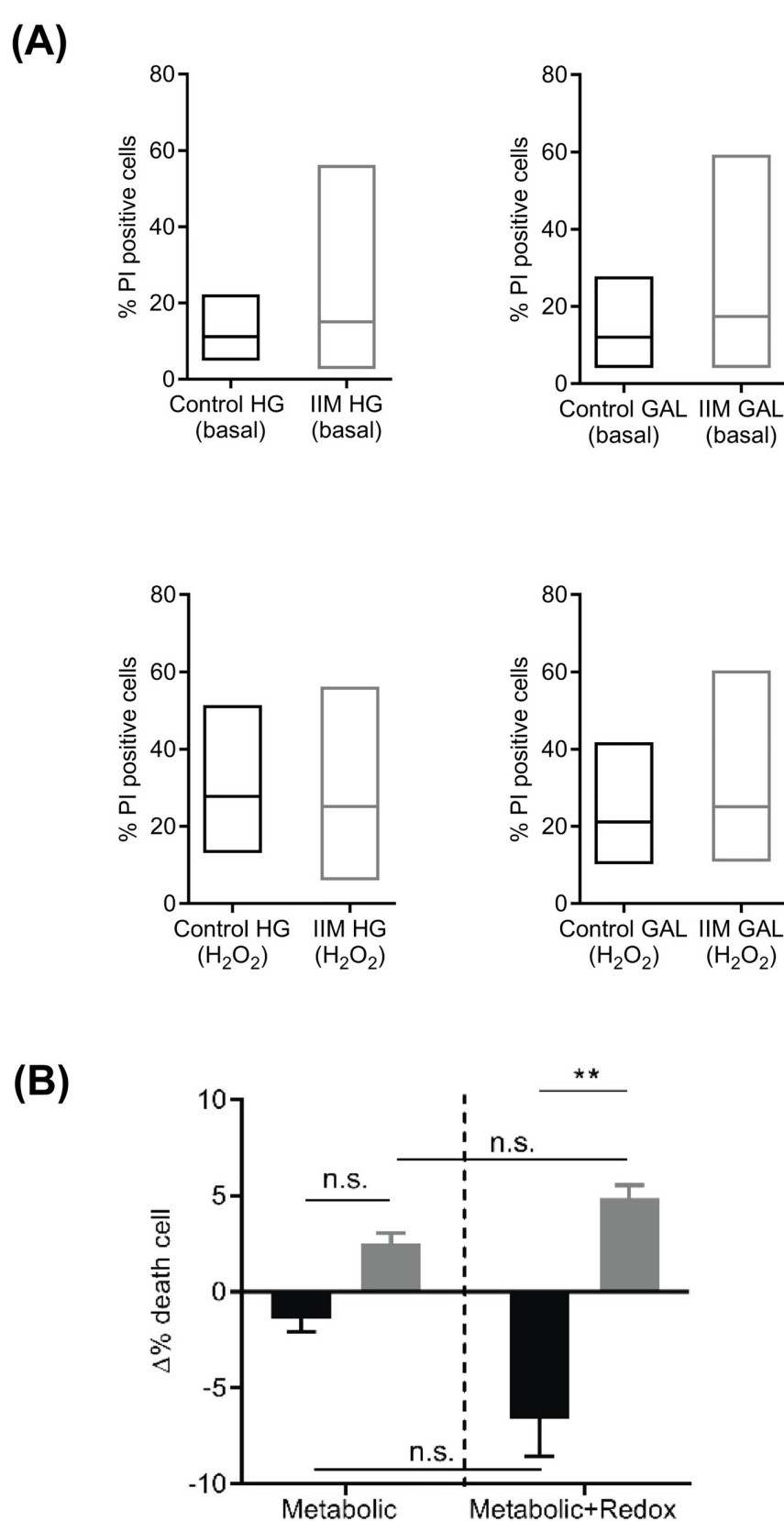

**Fig 3. IIM-derived cells are more susceptible to metabolic stress showing increased ROS-mediated cell death.**
Control and Idiopathic Inflammatory Myopathy (IIM)-derived cells were grown for seven days either in glucose (GLU) or galactose (GAL) media. Cell viability was measured with and without an acute redox stress induced by 100 μM $H_2O_2$. (A) IIM-derived cells reach the highest death values in every one of the four different conditions analyzed. The mean values for cell death also showed a tendency to be higher for the IIM condition, mainly in GLU media. (B) The delta of cell death (i.e. Δ = % death in galactose medium—% death glucose medium) was not significant for control and IIM-derived cells under "metabolic stress only". After an acute redox stress, the delta of cell death was significantly higher in IIM-derived cells than in controls. Data shown represent: (A) minimum, maximum and mean values of three independent experiments, (B) the mean ± SEM of three independent experiments: $^*p<0.05$, $^{**}p<0.01$, $^{***}p<0.001$, n.s. not significant.

calcium transfer to mitochondria [18], as well as an increase in the expression levels of nitric oxide synthase (NOS). As known, NOS is an enzyme that is highly related to increased ROS generation and necrosis [22]. Elevated ROS levels have been found in biopsy samples from DM patients. Likewise, higher levels of $H_2O_2$ have been described in skinned fibers [17]. In the current research, we observed an increase in ROS levels in IIM-derived cells in both GLU- and GAL-media (Fig 2E and 2F). However, to be sure about the role of ROS, further experiments using molecular and/or pharmacological intervention of the redox homeostatic system are necessary.

The potential role of "non-immune-mediated" cellular processes in the pathophysiology of IIM is a relatively new area of research. Mitochondrial alterations were considered early in the history of IIM pathophysiology; however, cellular research in this field has been insufficient until recently. Histochemical evidence performed mainly on DM and PM muscle tissue suggested mitochondrial abnormalities as a mechanism that acts on the pathophysiology of IIM [6–8]. Our observations in this study added to the earlier idea of damaged mitochondria in IIM. More precisely, our findings seemed to reveal that in IIM-derived cells, mitochondria is not dysfunctional and can still respond to metabolic stress (which in this particular case was the replacement of GLU by GAL) by increasing their respiration (Fig 2). This feature suggests that IIM-derived cells can still sense and transduce the availability of nutrients and energy to adapt to environmental changes. Previously, Robinson et al. (1992) took advantage of this kind of stress by testing the ability of patient's fibroblasts to survive a challenge, which consists of growing cells in galactose medium as the main source of carbon compounds [23]. They postulated this strategy as a test procedure to detect certain types of oxidative defects. Namely, they observed that cells with severe oxidative defects experienced increased susceptibility to death when cultured in galactose media as their sole source of carbohydrates. Similarly, in our research we tested that notion after allowing an adaptation time frame, studying whether IIM cells were more susceptible to dying in GLU or GAL media. As seen in Fig 3, IIM cells tended to die in higher proportions than controls as a result of metabolic stress. This trend became even more pronounced once a redox stress ($H_2O_2$) was applied. The IIM cell death rate was significantly increased for cells cultured in GAL media. We interpret this finding, which we believe to be one of the main contributions, as an increased susceptibility of IIM cells to perish when mitochondria were forced to increase their oxygen consumption. The aforementioned approach has also been used to test whether different toxic compounds can affect mitochondria in cultured cells. Dott et al. [24] demonstrated that mitochondria from L6 myoblasts cultured in galactose media were more susceptible to classic mitochondrial toxins. Furthermore, they demonstrated slower proliferation and increased OXPHOS capacity for L6 cells cultured in galactose media compared to cells cultured in glucose.

For the purpose of the present research, testing the mechanisms underlying the higher cell death rates of IIM cells grown in GAL medium was beyond our scope. Two recent studies that focused on IIM are worth mentioning. Both studies pointed to increased ROS levels as

significant contributors to the pathophysiology of the disease. Increased amounts of ROS (measured as DHE) in biopsy samples from DM patients, as well as higher $H_2O_2$ production in skinned fibers were described by Meyer et al. [17]. These researchers postulated a role for ROS in the pathophysiology of the disease, suggesting a direct damage to mitochondria, inducing their malfunction. In the same study, an experimental mouse model with autoimmune myositis confirmed increased ROS levels in quadriceps and gastrocnemius muscles. Supplementation of NAC (an antioxidant) to mice resulted in reduced ROS levels and preserved grip strength. A reduction in mitochondrial respiration was also observed in permeabilized fibers [17]. Lightfoot et al. [18] also proposed ROS as an important mediator in IIM pathophysiology. The review hypothesized about the mechanisms that could explain the increased ROS in IIM cells: they suggested that endoplasmic reticulum stress had the potential to increase calcium influx into mitochondria, thus increasing mitochondrial ROS production. Although our results also revealed increased ROS levels in IIM-derived cells (Fig 2E and 2F), we did not investigate ROS differences between GLU and GAL media. We recognize that in the absence of such a measure, we cannot hypothesize the existence of a possible relationship between the increase in ROS levels and the increase in the mortality rate in GAL media.

Our observations, made for the first time in primary human skeletal muscle cells, showed that mitochondria from IIM patients were able to exhibit plasticity, showing the capacity to increase their functional status when necessary (i.e. the replacement of glucose by galactose), as determined by oxygen consumption measurements. This "mitochondrial flexibility" seemed to suggest that this organelle is still capable of detecting and transducing environmental signals, in order to maintain the relation between mitochondrial function and cellular signaling. Nonetheless, the fact that higher death rates were observed for IIM cells suggested that the processes required for adaptation might ultimately be detrimental to the survival of cells. This finding shed light on the role that mitochondria might play in the weakness and atrophy seen in these patients.

In summary, our evidence suggests that boosting mitochondrial function in IIM, as is done in other muscle-related diseases, could have a detrimental effect on skeletal muscle health. Future research should focus on unraveling the balance between ROS generation and ROS scavenging to develop new therapeutic strategies for this complex group of diseases.

## 5. Conclusion

Mitochondria of IIM-derived cells are functional and can adapt and respond to a metabolic challenge. However, a concomitant effect is that cells tend to become more susceptible to cellular insults that result in increased death.

## Supporting information

**S1 Fig. Mitochondrial content is similar between control and IIM patients.** (A) Human skeletal muscle biopsies obtained from controls and IIM patients were labeled with a specific antibody against the outer membrane mitochondrial protein VDAC. Equally sized "Regions Of Interest" (ROIs) were analyzed with Image J, and the area was expressed in pixel units. No differences were observed between controls and IIM patients. Controls n = 3; Patients n = 4. (B). Mitochondrial complexes (I, III, IV and V) were analyzed by Western Blot in tissue samples from human skeletal muscle biopsies obtained from control and IIM patients. No differences were observed. Controls n = 3; IIM n = 3. Mean ± SEM.
(TIF)

## Acknowledgments

Thanks to Daniel Rappaport (M.D.), for helping with biopsy procedures and to Samuel Navarro Ortega, PhD and Alenka Lovy, PhD for proof-reading the manuscript.

## Author Contributions

**Conceptualization:** Carla Basualto-Alarcón, Félix A. Urra, María Francisca Bozán, Jorge A. Bevilacqua, J. César Cárdenas.

**Data curation:** Carla Basualto-Alarcón, Félix A. Urra, Fabián Jaña, Alejandra Trangulao, J. César Cárdenas.

**Formal analysis:** Carla Basualto-Alarcón, Félix A. Urra, Jorge A. Bevilacqua, J. César Cárdenas.

**Funding acquisition:** Jorge A. Bevilacqua, J. César Cárdenas.

**Investigation:** Carla Basualto-Alarcón, Fabián Jaña, J. César Cárdenas.

**Methodology:** Carla Basualto-Alarcón.

**Project administration:** J. César Cárdenas.

**Resources:** María Francisca Bozán, Jorge A. Bevilacqua, J. César Cárdenas.

**Supervision:** J. César Cárdenas.

**Writing – original draft:** Carla Basualto-Alarcón, J. César Cárdenas.

**Writing – review & editing:** J. César Cárdenas.

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
