## [Decision Letter · Decision Letter 0]

17 Jul 2020

PONE-D-20-20169

Idiopathic Inflammatory Myopathy Human Derivate Cells retain their ability to increase mitochondrial function

PLOS ONE

Dear Dr. Cardenas,

Thank you for submitting your manuscript to PLOS ONE. After careful consideration, we feel that it has merit but does not fully meet PLOS ONE’s publication criteria as it currently stands. Therefore, we invite you to submit a revised version of the manuscript that addresses the points raised during the review process

We look forward to receiving your revised manuscript.

Kind regards,

Jianhua Zhang

Academic Editor

PLOS ONE

Journal Requirements:

Reviewers' comments:

Reviewer's Responses to Questions

**Comments to the Author**

1. Is the manuscript technically sound, and do the data support the conclusions?

Reviewer #1: Partly

Reviewer #2: Yes

2. Has the statistical analysis been performed appropriately and rigorously? 

Reviewer #1: Yes

Reviewer #2: Yes

3. Have the authors made all data underlying the findings in their manuscript fully available?

Reviewer #1: Yes

Reviewer #2: Yes

4. Is the manuscript presented in an intelligible fashion and written in standard English?

Reviewer #1: Yes

Reviewer #2: Yes

5. Review Comments to the Author

Reviewer #1: The manuscript, “Idiopathic Inflammatory Myopathy Derivative Cells retain their ability to increase mitochondrial function” by Basualto-Alarcon et al. investigates mitochondrial function, ATP production, and ROS generation in myoblasts derived from patients suffering from some form of idiopathic inflammatory myopathy (IIM), comparing them to healthy controls. The authors demonstrate that both basal and ATP-linked oxygen consumption rates (OCR) are decreased in IIM myoblasts, which is associated with uncoupled, hyperpolarized mitochondria. Furthermore, the authors show that changing the growth conditions of IIM myoblasts from high glucose to galactose can rescue the defects in mitochondrial function; however, these myoblasts are more susceptible to cell death. The strongest aspect of this study is the translational relevance of using human myoblasts from patients actually suffering from these diseases. However, the novelty is a bit lacking due to the previously cited publications (References 15 and 16) indicating galactose improves mitochondrial function in differentiated myoblasts from diabetic patients and C2C12 mouse myoblasts, respectively. While the possible translational importance is high, there are a number of instances where the authors overstate some results based on the data provided. There is also some contradiction between how the results are interpreted and then covered in the discussion. Suggestions to improve are included below.

1) Realistically, this work should all be done in differentiated myoblasts, or at least the authors could give a comparison between undifferentiated and differentiated myoblasts. Presumably, the undifferentiated cells are much more glycolytic (as the authors cover regarding the Crabtree effect in cell culture with high glucose), and once differentiated would rely on mitochondrial respiration more. At that point, the authors might see more significant effects with galactose supplementation (similar to Ref 15) than they did with the undifferentiated cells (i.e. less death, ATP differences, ROS changes).

2) The authors argue that the myoblasts are more or less glycolytic without ever showing any measure of glycolysis. As the Seahorse was used for OCR measurements, the authors could also include ECAR graphs to show changes in glycolytic flux in the presence of glucose or galactose in control versus IIM myoblasts. The utilization of 2-DG, or even the seahorse glycolytic stress test (Glucose/Galactose-Oligomycin-2DG), coupled with the OCR data would provide a great deal of information regarding how glycolysis is being affected in these cells.

3) The results and discussion seem to disagree a bit regarding the actual functional capacity of IIM myoblast mitochondria. In the results, the argument seems to be that mitochondria in IIM cells retain their ability to function during stress but can only do so in the presence of galactose where they are forced to not rely on glycolysis. However, in the discussion, as well as the fact that the IIM myoblasts are more prone to die during metabolic/redox stress, the authors indicate that when IIM cells are forced to utilize their mitochondria, it enhances their susceptibility to cell death. Thus, it would seem that regardless of substrate, mitochondria of IIM patients are severely compromised, which should be a more prevalent focal point of this study.

4) Along the lines of point 3, the inclusion of some indicator of mitochondrial number/mass would be useful (i.e. mtDNA to nuclear DNA ratio, citrate synthase, MitoTracker/TOMM20 staining). All of the OCR is normalized to cell number, so it would be good to show whether or not the number of mitochondria is altered in IIM myoblasts. Regarding normalization, it might be better to normalize the OCR measurements to total protein or some other better indicator of final cell number post-assay. The authors mention in the methods that 20,000 cells were plated, but the graphs are all normalized to 106 cells, how was this number determined? This should be detailed in the materials and methods section. Similar to this point, do the IIM myoblasts grow at the same rate as the normal healthy controls? Based on their decreased metabolic capacity, it would seem possible that they grow slower and thus total cell number during the assay may be lower than control. Inclusion of cell counts or MTT data might solidify this point.

5) Figure 1 – The authors should include the actual BOFA curves as well as the bar graphs for Panel A. Also, why is the non-mitochondrial oxygen consumption rate so high in both the control and IIM myoblasts? Most OCR curves in the literature usually have relatively low non-mitochondrial contribution.

6) Figure 2 – Would the fact that the IIM myoblasts produce the same amount of ROS regardless of whether or not they are grown in glucose or galactose-containing media not indicate that the ROS is coming from non-mitochondrial sources? This possibility is briefly mentioned in the discussion but may confound the interpretation that the increased metabolic capacity afforded by galactose is increasing ROS levels due to increased mitochondrial function. Perhaps some indicator of the source of ROS (i.e. changes in MitoSox fluorescence) would help tease out where ROS is generated in the different conditions. Also, did the authors measure membrane potential with TMRM in the presence of galactose? It would be interesting to see if the hyperpolarization observed in IIM cells is decreased upon the provision of galactose (since the cells are more prone to death but have rescued OCR).

7) Figure 3 – While the interpretation of viability as a percent difference between media conditions is an interesting way to represent the data, it makes it a bit difficult to ascertain how many cells are actually alive versus dead. The authors should include a standard viability graph (i.e. %death or %viable cells) for each condition, then could include the assessment of percent change based on glucose or galactose.

8) The discussion could use some editing for proper English.

Reviewer #2: The study by Basualto-Alarcon examined mitochondrial function and adaptation in skeletal muscle-derived cells from idiopathic inflammatory myopathy patients during metabolic challenges, in this study modeled through glucose deprivation. Their conclusion is well supported by the data and the limitations of the study well raised and argued in the discussion section.

Therefore I have no further comment

6. PLOS authors have the option to publish the peer review history of their article (what does this mean?). If published, this will include your full peer review and any attached files.

Reviewer #1: No

Reviewer #2: No

---

## [Author Response · Author response to Decision Letter 0]

4 Sep 2020

Regarding our work entitled “Idiopathic Inflammatory Myopathy Human Derived Cells retain their ability to increase mitochondrial function” (PONE-D-20-20169) we thank the reviewers for their comments and the opportunity to improve our manuscript. We have revised the manuscript and addressed the issues that have been raised to the best of our abilities under the difficulties imposed by the COVID-19 pandemia.

Reviewer comments: 

Reviewer #1: The manuscript, “Idiopathic Inflammatory Myopathy Derivative Cells retain their ability to increase mitochondrial function” by Basualto-Alarcón et al. investigates mitochondrial function, ATP production, and ROS generation in myoblasts derived from patients suffering from some form of idiopathic inflammatory myopathy (IIM), comparing them to healthy controls. The authors demonstrate that both basal and ATP-linked oxygen consumption rates (OCR) are decreased in IIM myoblasts, which is associated with uncoupled, hyperpolarized mitochondria. Furthermore, the authors show that changing the growth conditions of IIM myoblasts from high glucose to galactose can rescue the defects in mitochondrial function; however, these myoblasts are more susceptible to cell death. The strongest aspect of this study is the translational relevance of using human myoblasts from patients actually suffering from these diseases. However, the novelty is a bit lacking due to the previously cited publications (References 15 and 16) indicating galactose improves mitochondrial function in differentiated myoblasts from diabetic patients and C2C12 mouse myoblasts, respectively. While the possible translational importance is high, there are a number of instances where the authors overstate some results based on the data provided. There is also some contradiction between how the results are interpreted and then covered in the discussion. Suggestions to improve are included below.

Response: We thank the reviewers for acknowledging the strength of our work. Certainly, it was a challenge to work with patient derived samples. 

Based on feedback from reviewer #1, we found that we made the mistake of not clearly communicating our main observation, which is not that galactose improves mitochondrial function, but that, independent of the growing medium, IIM mitochondria are able to sense and adapt to environmental stress at the cost that renders IIM cells more prone to suffer oxidative damage and to die. Until today, all the publications regarding mitochondria in IIM suggest mitochondrial dysfunction, which is actually not the case.

1) Realistically, this work should all be done in differentiated myoblasts, or at least the authors could give a comparison between undifferentiated and differentiated myoblasts. Presumably, the undifferentiated cells are much more glycolytic (as the authors cover regarding the Crabtree effect in cell culture with high glucose), and once differentiated would rely on mitochondrial respiration more. At that point, the authors might see more significant effects with galactose supplementation (similar to Ref 15) than they did with the undifferentiated cells (i.e. less death, ATP differences, ROS changes).

Response: We agree with the reviewer that it would be interesting to compare myoblast and myotubes. Unfortunately, in our hands, the differentiation of normal and IIM derivative cells was extremely slow, which made it cost-inefficient and impossible for us to maintain in the long run. Thus, we decided to perform the experiments on myoblasts in the light of previous work that shows that myoblast and myotubes present similar respiration patterns (Olah et al., 2015, Suman et al., 2018, Hoffmann et al., 2018). Thus, we believe similar results will be expected in myotubes and the conclusions will remain the same. We present here some of aformentioned evidence:

● Olah et al. PLoS ONE 10(7):e0134227. 2015. (doi:10.1371/journal.pone.0134227).

● Suman et al. Human Molecular Genetics, 27:2367 – 2382. 2018

(doi:10.1093/hmg/ddy149).

● Hoffmann et al. Scientific Report 8: 737. 2018. (doi.org/10.1038/s41598-017-18658-3).

2) The authors argue that the myoblasts are more or less glycolytic without ever showing any measure of glycolysis. As the Seahorse was used for OCR measurements, the authors could also include ECAR graphs to show changes in glycolytic flux in the presence of glucose or galactose in control versus IIM myoblasts. The utilization of 2-DG, or even the seahorse glycolytic stress test (Glucose/Galactose-Oligomycin-2DG), coupled with the OCR data would provide a great deal of information regarding how glycolysis is being affected in these cells.

Response: We apologize for our writing style, which may have led to a confusion about our experiment’s interpretation. Regarding a hypothetical glycolytic derived ATP production in control and IIM cells in high glucose conditions, we proposed this as a possible explanation, in light of our results showing that oligomycin addition, didn’t diminish “total ATP levels (figure 1C)”. As reviewer #1 suggests, the glycolytic Seahorse stress test would have contributed to clarify this point, however, at the time we discarded to perform this experiment as it was not our main objective. Right now, given the current global epidemic situation, we were not allowed to go in the lab to perform the suggested experiments. Nevertheless, we had gone through the manuscript and changed some of our expressions, as well as added a sentence explaining our interpretation. These changes are in red in the new version of the manuscript.

The new sentence reads as follows (the additional wording is in red):

“This result suggests that in standard, high glucose culture conditions, both control and IIM-derived cells could rely mostly on glycolysis to synthesize their ATP, since oligomycin, an inhibitor of mitochondrial ATP synthase, did not decrease ATP levels as expected for a cell that produces its ATP through the oxidative pathway.”

The ECAR measurement done in parallel with the OCR does not show differences between Control and IIM cells, but we know that this is inconclusive and the glycolytic seahorse stress maneuver is the one that will reveal the detail of the glycolytic function in these cells. Nevertheless, we included here one typical ECAR graph obtained from our measurement.

3) The results and discussion seem to disagree a bit regarding the actual functional capacity of IIM myoblast mitochondria. In the results, the argument seems to be that mitochondria in IIM cells retain their ability to function during stress but can only do so in the presence of galactose where they are forced to not rely on glycolysis. However, in the discussion, as well as the fact that the IIM myoblasts are more prone to die during metabolic/redox stress, the authors indicate that when IIM cells are forced to utilize their mitochondria, it enhances their susceptibility to cell death. Thus, it would seem that regardless of substrate, mitochondria of IIM patients are severely compromised, which should be a more prevalent focal point of this study.

Response: We agree with the reviewer that there seems to be a disconnection between the description of the results section and the discussion regarding the status of mitochondria in IIM. The main take home message from our work is that mitochondria in IIM show less respiration not because they are dysfunctional, but because they adapt their function to improve survival. This adaptation was experimentally revealed by changing glucose for galactose. We amended the manuscript and reinforced this point throughout the discussion.

Results section:

Taken together, these results suggest that mitochondria retain the potential to reach full functionality in IIM cells independently of the growth culture medium, and that the decrease we observed in mitochondrial function in GLU-media may correspond to an adaptive response undergoing the aforementioned “Crabtree effect”.

Discussion section: 

It has been proposed that in IIM, the inflammatory local response is not the only factor responsible for the pathophysiology of this illness (2). There is evidence that points to mitochondrial malfunction also being responsible, proposing that mitochondria are more likely to produce high levels of ROS and a decrease in ATP synthesis capacity (2-12, 18). In this work, we show for the first time that in human derived IIM myoblasts, mitochondria are not dysfunctional and retain their ability to adapt and respond to environmental signals, at a cost that renders cells more prone to death after a specific, oxidative, insult. 

Our observations in this study corroborated and added to the earlier idea of damaged mitochondria in IIM. More precisely, our findings seemed to reveal that in IIM-derived cells, mitochondria could still respond to metabolic stress (which in this particular case was the replacement of GLU by GAL) by increasing their respiration (fig. 2). This feature suggests that IIM-derived cells can still sense and transduce the availability of nutrients and energy to adapt to environmental changes. 

The IIM cell death rate was significantly increased for cells cultured in GAL media. We interpret this finding, which we believe to be one of the main contributions, as an increased susceptibility of IIM cells to perish when mitochondria were forced to increase their oxygen consumption. 

Our observations, made for the first time in primary human skeletal muscle cells, showed that mitochondria from IIM patients were able to exhibit plasticity, showing the capacity to increase their functional status when necessary (i.e. the replacement of glucose by galactose), as determined by oxygen consumption measures. 

4) Along the lines of point 3, the inclusion of some indicator of mitochondrial number/mass would be useful (i.e. mtDNA to nuclear DNA ratio, citrate synthase, MitoTracker/TOMM20 staining). All of the OCR is normalized to cell number, so it would be good to show whether or not the number of mitochondria is altered in IIM myoblasts.

Response: We agree with the reviewer that information regarding mitochondrial mass in control and IIM cells would add interesting data, as well as allow a better analysis of our functional experiments. We conducted different strategies directed to measure mitochondrial mass. Control and IIM myoblasts were labeled with 10-N-Nonyl acridine orange (NAO), which had been extensively used to determine mitochondrial mass thanks to its ability to bind cardiolipin (Septinus M, et al., 1985). As shown in the new figure 2D, no NAO label differences were observed between control and IIM myoblasts in galactose medium, which suggests no mitochondrial mass differences. 

In addition, patient and control biopsies were labeled with the specific outer mitochondrial membrane protein, VDAC and the pixel intensity analyzed. As shown here for the reviewers, no differences were observed between control and IIM biopsies, regarding pixel intensities. This particular data is part of a different manuscript which is the reason we are not including it in this paper, but we share it. 

Septinus et al., Histochemistry. 1985; 82(1):51-66)

Patients and controls did not differ for VDAC

area. Biopsies from controls and patients were

analyzed by immunofluorescence for VDAC channel. Equally sized "Regions Of Interest” (ROIs) were analyzed with Image J, and the area was expressed in pixel units.

No differences were observed for controls and patients. Controls n=3; Patients n=4. 

4.1) Regarding normalization, it might be better to normalize the OCR measurements to total protein or some other better indicator of final cell number post-assay. The authors mention in the methods that 20,000 cells were plated, but the graphs are all normalized to 106 cells, how was this number determined? This should be detailed in the materials and methods section. 

Response: As the reviewer points out, we seeded 20,000 cells per well for Seahorse experiments. Although, we decided to normalize our results, by a value of 106 cells, for the following reasons:

- Normalizing by 106 cells is a way to standardize the OCR measurements along with the respiration measurements obtained by other techniques of respiration, i.e. Clark-type electrode. Those experiments use a large number of cells, and are normalized by a million of cells (106 cells), thus, in an attempt to homogenize the numbers of our experimental system, we multiplied the numbers obtained from 20,000 cells by a 50-factor. In this way, our results can be easily compared with other data from other equipment.

- As different cell types may have a different optimal number for the “number of seeded cells” to measure OCR, then having a number that allows normalization, helps us to compare different cell types, under different conditions. 

- Moreover, in our laboratory we have previously described the changes in mitochondrial respiration using XFe96 technology expressing the data in pmol O2/min/106 cells in previous works (for example, supplementary figure 7g, Urra et al. 2018. 

 Urra et al., 2018, Scientific Report; 8(1):13190.

4.2) Similar to this point, do the IIM myoblasts grow at the same rate as the normal healthy controls? Based on their decreased metabolic capacity, it would seem possible that they grow slower and thus total cell number during the assay may be lower than control. Inclusion of cell counts or MTT data might solidify this point.

Response: This is a good point. In order to avoid this bias, oxygen consumption measures were always done 12 hours after seeding the cells, which is a time frame in which neither control nor IIM cells are able to divide. 

5) Figure 1 – The authors should include the actual BOFA curves as well as the bar graphs for Panel A. Also, why is the non-mitochondrial oxygen consumption rate so high in both the control and IIM myoblasts? Most OCR curves in the literature usually have relatively low non-mitochondrial contribution.

Response: We thank reviewer # 1 for his/her suggestion. We incorporated this new graph in figure 1, which now shows a representative OCR profile for control and IIM cells (fig 1A). 

Regarding the non-mitochondrial OCR, it is true that the values are higher than those we usually observe in our experiments, but is not totally rare. We find in the literature a work in primary old and young skeletal muscle cells where a high non-mitochondrial OCR is observed (Pala et al., 2018). We believe this is caused by a high cellular metabolism and as it is present in both control and IIM cells we don’t expect this to change/modify the main conclusions of our work. Also, there is a small possibility that the aliquot of rotenone and antimycin A (AA) we used was not in perfect conditions, causing only a partial inhibition. Again, this will not change the main conclusions of our work.

Pala et al., 2018; Journal of Cell Science 131(14):jcs212977.

6) Figure 2 – Would the fact that the IIM myoblasts produce the same amount of ROS regardless of whether or not they are grown in glucose or galactose-containing media not indicate that the ROS is coming from non-mitochondrial sources? This possibility is briefly mentioned in the discussion but may confound the interpretation that the increased metabolic capacity afforded by galactose is increasing ROS levels due to increased mitochondrial function. Perhaps some indicator of the source of ROS (i.e. changes in MitoSox fluorescence) would help tease out where ROS is generated in the different conditions. Also, did the authors measure membrane potential with TMRM in the presence of galactose? It would be interesting to see if the hyperpolarization observed in IIM cells is decreased upon the provision of galactose (since the cells are more prone to death but have rescued OCR).

Response: This is a very interesting observation made by the reviewer. Unfortunately, our ROS measurements were done in separate time frames. This makes it impossible to compare our glucose-containing media data with our galactose containing-media data. It would be good to perform these experiments in parallel and compare, but unfortunately we cannot go to the lab due to the covid-19 situation. We recognize this as a weakness of our work and we mention this in the discussion section. Also, measuring the mitochondrial membrane potential is a great suggestion that we cannot perform right now. Nevertheless, we believe our work is still very relevant information to the scientific community. 

7) Figure 3 – While the interpretation of viability as a percent difference between media conditions is an interesting way to represent the data, it makes it a bit difficult to ascertain how many cells are actually alive versus dead. The authors should include a standard viability graph (i.e. %death or %viable cells) for each condition, then could include the assessment of percent change based on glucose or galactose.

Response: We have added a new figure 3A to show the results in an easier way. We kept the original figure as figure 3B.

8) The discussion could use some editing for proper English.

Response: We are sorry for any inappropriate use of the English language. A native English speaker has looked over and corrected our work.

Reviewer #2: The study by Basualto-Alarcon examined mitochondrial function and adaptation in skeletal muscle-derived cells from idiopathic inflammatory myopathy patients during metabolic challenges, in this study modeled through glucose deprivation. Their conclusion is well supported by the data and the limitations of the study well raised and argued in the discussion section.

Therefore I have no further comment

Response: We appreciate the reviewer’s comments.

---

## [Decision Letter · Decision Letter 1]

22 Sep 2020

PONE-D-20-20169R1

Idiopathic Inflammatory Myopathy Human Derived Cells retain their ability to increase mitochondrial function

PLOS ONE

Dear Dr. Cardenas

Thank you for submitting your manuscript to PLOS ONE. After careful consideration, we feel that it has merit but does not fully meet PLOS ONE’s publication criteria as it currently stands. Therefore, we invite you to submit a revised version of the manuscript that addresses the points raised during the review process.

We look forward to receiving your revised manuscript.

Kind regards,

Jianhua Zhang

Academic Editor

PLOS ONE

Reviewers' comments:

Reviewer's Responses to Questions

**Comments to the Author**

1. If the authors have adequately addressed your comments raised in a previous round of review and you feel that this manuscript is now acceptable for publication, you may indicate that here to bypass the “Comments to the Author” section, enter your conflict of interest statement in the “Confidential to Editor” section, and submit your "Accept" recommendation.

Reviewer #1: All comments have been addressed

Reviewer #2: (No Response)

2. Is the manuscript technically sound, and do the data support the conclusions?

Reviewer #1: Yes

Reviewer #2: Partly

3. Has the statistical analysis been performed appropriately and rigorously? 

Reviewer #1: Yes

Reviewer #2: Yes

4. Have the authors made all data underlying the findings in their manuscript fully available?

Reviewer #1: Yes

Reviewer #2: Yes

5. Is the manuscript presented in an intelligible fashion and written in standard English?

Reviewer #1: Yes

Reviewer #2: Yes

6. Review Comments to the Author

Reviewer #1: (No Response)

Reviewer #2: The authors have addressed major concerns of the manuscript and have thus improved the quality of the paper, however, a few points still need to be reviewed.

1- The authors have provided data to show that mitochondrial mass is similar between both Controls and IMM using NAO probe. Using this probe, the authors have to mitigate their interpretation of the data for a few reasons which are:

-NAO accumulates in mitochondria in a potential-dependent manner. The authors having stated that mitochondria are hyperpolarized in IMM, there's a high possibility that the dye distribution is different between CTL and IMM.

-Cardiolipin are also found in peroxisomes

I suggest the authors use other indicators of mitochondrial mass such as mtdna/nuclear dna ratio in all conditions (glucose and galactose). Indeed, several studies had shown an increase in mitochondrial turnover in galactose, I believe the paper would be greatly improved to show that the observed changes are not attributed to changes in mitochondrial mass.

2-Authors suggest that forcing IMM cells to rely on mitochondrial synthesized ATP renders the cells more prone to death following acute Oxidative insult. The study would be strengthened by inhibition of ROS production, to further establish a direct role of ROS production in the IMM cells sensitivity to death, particularly in galactose medium culture.

3-Authors have shown graph of ECAR measurement which do not show differences between Control and IMM. I believe these were done in high glucose medium. It would be interesting to show these ECAR measurements in Galactose medium.

7. PLOS authors have the option to publish the peer review history of their article (what does this mean?). If published, this will include your full peer review and any attached files.

Reviewer #1: No

Reviewer #2: No

---

## [Author Response · Author response to Decision Letter 1]

22 Oct 2020

Responses to Reviewers’ Comments

Regarding our work entitled “Idiopathic Inflammatory Myopathy Human Derivate Cells retain their ability to increase mitochondrial function” (PONE-D-20-20169) we want to thank reviewer 1 for accepting our amended manuscript and reviewer 2 for acknowledging that the quality of our work has greatly improved and that all his major concerns have been addressed. 

Reviewer comments: 

Reviewer #2: The authors have addressed major concerns of the manuscript and have thus improved the quality of the paper, however, a few points still need to be reviewed.

1- The authors have provided data to show that mitochondrial mass is similar between both Controls and IMM using NAO probe. Using this probe, the authors have to mitigate their interpretation of the data for a few reasons which are:

-NAO accumulates in mitochondria in a potential-dependent manner. The authors having stated that mitochondria are hyperpolarized in IMM, there's a high possibility that the dye distribution is different between CTL and IMM.

-Cardiolipin are also found in peroxisomes

I suggest the authors use other indicators of mitochondrial mass such as mtdna/nuclear dna ratio in all conditions (glucose and galactose). Indeed, several studies had shown an increase in mitochondrial turnover in galactose, I believe the paper would be greatly improved to show that the observed changes are not attributed to changes in mitochondrial mass.

Response: We thank the reviewer for this insightful comment. We agree that NAO is not the best probe to determine mitochondrial mass. Conflicted results have been published regarding its use (REF, siy no). Related to our experiments, the fact that the measurements with NAO are similar in control and IMM cells, despite IMM cells being hyperpolarized may suggest that; 1- IMM cells have way less mitochondria but given the high mitochondrial membrane potential, accumulated so much dye that the quantification is similar to control, or 2- NAO binds mainly to cardiolipin and there is no difference in mitochondrial mass. 

As mentioned by the reviewer the best methods to quantify mitochondrial mass is by performing a mitochondrial DNA/nuclear DNA ratio. This is a technique that we want to implement in our lab but unfortunately is not working at the moment. In the current situation, we have no access to patients’ samples to culture cells and do the required experiments. All the health personnel is working on covid-19 related tasks and all other projects are on hold. We have some frozen material, but the cells were not able to grow. The only material we have available are biopsy samples. We performed Western blots of the mitochondrial complexes in biopsy samples from normal and IIM patients. In agreement with the previous analysis of VDAC performed on biopsy samples from control and IIM patients, no differences in the expression of complex I, III, IV and V were found between the samples. This data has been added as supporting figure 1. Altogether, our data suggest that no difference in mitochondrial mass is observed between normal and IIM conditions, however, we understand that better experiments can be done to show this with less doubt. We made a statement regarding this in the discussion. We add the following sentences (red) to the results and the discussion section;

Results

…” Nonylacridine orange (NAO) staining show no differences between control and IIM cells (fig. 2D) suggesting no differences in mitochondrial mass. To strengthen this point, we labeled biopsy samples from control and IIM patients with an antibody against the outer mitochondrial protein VDAC and we determined its expression by quantifying the number of pixels per area in confocal microscope images. As shown in the supporting figure 1A, no difference in the distribution nor in the expression were found between normal and IIM samples. In addition, using an antibody cocktail we determined the expression of the electron transport chain complexes by Western blot in biopsy samples from control and IIM patients. As shown in supporting figure 1B, no changes in the expression of complex I, III, IV and V were observed between control and IIM samples. Due to technical reasons complex II was not identified in any of our samples. These results strengthen the idea that no difference in mitochondrial mass exists between normal and IIM cells. More experiments are necessary to confirm this point.

Discussion

…” We interpret this result as a demonstration of the IIM cell ability to adapt successfully to a metabolic stress condition. Nonylacridine orange (NAO) labeling of cells in culture, in addition to the labeling of the outer mitochondrial protein VDAC and Western blot of the electron transport chain complexes I, III, IV and V on biopsy samples from control and IIM patients, suggests that no changes in mitochondrial mass between normal and IIM samples exist. NAO binds cardiolipin, which is highly concentrated in the mitochondria independent of ΔΨm (19), however under certain circumstances it accumulates in the mitochondria in a ΔΨm-dependent fashion (20,21). To confirm that the mitochondrial mass is similar between control and IIM samples, further experiments measuring the mitochondrial/nuclear DNA ratio are necessary.

2-Authors suggest that forcing IMM cells to rely on mitochondrial synthesized ATP renders the cells more prone to death ffollowing acute Oxidative insult. The study would be strengthened by inhibition of ROS production, to further establish a direct role of ROS production in the IMM cells sensitivity to death, particularly in galactose medium culture.

Response: We agree with the reviewer. The utilization of antioxidants or other means to inhibit ROS would strengthen our conclusion. Unfortunately, we don’t have cells to perform this experiment and given the covid-19 pandemic, all the health personnel that could help us with the samples, including Dr Bozan, are now working to contain covid-19. In the discussion we have stated that experiments with antioxidants are necessary to prove the role of ROS in the cell death observed.

…” In the current research, we observed an increase in ROS levels in IIM-derived cells in both GLU- and GAL-media (fig. 2E and 2F). However, to be sure about the role of ROS, further experiments using molecular and/or pharmacological intervention of the redox homeostatic system are necessary.

3-Authors have shown graph of ECAR measurement which do not show differences between Control and IMM. I believe these were done in high glucose medium. It would be interesting to show these ECAR measurements in Galactose medium.

Response: As mentioned by the reviewer, the ECAR measurements we share with the reviewers in our first respond were indeed performed in high glucose. Here we added a representative ECAR of cells growth in galactose. As shown in the figure 1 of this respond, no differences in ECAR are observed between control and IIM cells grown in galactose medium. Further experiments using the glycolysis stress kit are necessary to get more details about glycolysis in these cells. However, the lack of differences observed in these measurements suggest that no differences will be found. 

Nevertheless, the main focus of this manuscript is on the mitochondrial function and its unexpected deleterious effect on IIM cells. 

We understand that we have not been able to fully respond to the observation of reviewer 2, not because we didn’t consider his/her input, but because of the catastrophe caused by covid-19, which has basically stopped all research in our country. The main substrate for our research are the samples from patients, to which we have no access now. All health personnel, including authors of this manuscript are now fighting covid-19 and non-medical personnel are not considered necessary and therefore not allowed access. The basic researchers involved in this project cannot go to the hospital. We have tried to respond to best of our ability. We also modified the discussion to reveal the weakness of our work. Nevertheless, as is, we believe our work offers new insight into a problem poorly studied and Plos One will give us the exposure we need.

---

## [Decision Letter · Decision Letter 2]

3 Nov 2020

Idiopathic Inflammatory Myopathy Human Derived Cells retain their ability to increase mitochondrial function

PONE-D-20-20169R2

Dear Dr. Cardenas

We’re pleased to inform you that your manuscript has been judged scientifically suitable for publication and will be formally accepted for publication once it meets all outstanding technical requirements.

Kind regards,

Jianhua Zhang

Academic Editor

PLOS ONE

Additional Editor Comments (optional):

Reviewers' comments:

Reviewer's Responses to Questions

**Comments to the Author**

1. If the authors have adequately addressed your comments raised in a previous round of review and you feel that this manuscript is now acceptable for publication, you may indicate that here to bypass the “Comments to the Author” section, enter your conflict of interest statement in the “Confidential to Editor” section, and submit your "Accept" recommendation.

Reviewer #2: All comments have been addressed

2. Is the manuscript technically sound, and do the data support the conclusions?

Reviewer #2: No

3. Has the statistical analysis been performed appropriately and rigorously? 

Reviewer #2: Yes

4. Have the authors made all data underlying the findings in their manuscript fully available?

Reviewer #2: (No Response)

5. Is the manuscript presented in an intelligible fashion and written in standard English?

Reviewer #2: Yes

6. Review Comments to the Author

Reviewer #2: The authors have addressed all major concerns of the manuscript. Authors have amended the discussion to include the limitations of some of the techniques. The paper is acceptable for publication.

7. PLOS authors have the option to publish the peer review history of their article (what does this mean?). If published, this will include your full peer review and any attached files.

Reviewer #2: No

---

## [Editor Report · Acceptance letter]

6 Nov 2020

PONE-D-20-20169R2 

Idiopathic Inflammatory Myopathy Human Derived Cells retain their ability to increase mitochondrial function 

Dear Dr. Cardenas:

I'm pleased to inform you that your manuscript has been deemed suitable for publication in PLOS ONE. Congratulations! Your manuscript is now with our production department. 

Kind regards, 

on behalf of

Dr Jianhua Zhang 

Academic Editor

PLOS ONE